# A Contrast-Enhanced CT-Based Deep Learning System for Preoperative Prediction of Colorectal Cancer Staging and RAS Mutation

**DOI:** 10.3390/cancers15184497

**Published:** 2023-09-10

**Authors:** Na Lu, Xiao Guan, Jianguo Zhu, Yuan Li, Jianping Zhang

**Affiliations:** 1Department of General Surgery, The Second Affiliated Hospital of Nanjing Medical University, No. 121, Jiangjiayuan Road, Nanjing 210011, Chinaxiaoguan@stu.njmu.edu.cn (X.G.); 2Department of Radiology, The Second Affiliated Hospital of Nanjing Medical University, Nanjing 210011, China; zhujianguo@njmu.edu.cn; 3Key Laboratory of Modern Toxicology, Ministry of Education, School of Public Health, Nanjing Medical University, Nanjing 211166, China; liyuan@njmu.edu.cn

**Keywords:** colorectal cancer, stage, RAS status, deep learning, convolutional neural networks, transformer

## Abstract

**Simple Summary:**

This study explored the role of CT-based deep learning in detecting colorectal cancer tumor location and preoperatively predicting the stage and RAS gene mutation status of colorectal cancer patients. The deep learning model we built achieved excellent performance. The detection network based on Yolov7 realized the detection and preoperative staging of colorectal cancer with an average mean accuracy of 0.98 in the validation cohort. The vision transformer-based prediction network achieved accurate prediction of preoperative RAS in colorectal cancer patients, achieving an area under the receiver operating characteristic curve (AUC) of 0.9591 and 0.9554 in the test cohort and the validation cohort, respectively. This study also explored the clinical applications of deep learning models. Based on the proposed detection network and prediction network, we built a deep learning system for clinicians who do not understand deep learning.

**Abstract:**

Purpose: This study aimed to build a deep learning system using enhanced computed tomography (CT) portal-phase images for predicting colorectal cancer patients’ preoperative staging and RAS gene mutation status. Methods: The contrast-enhanced CT image dataset comprises the CT portal-phase images from a retrospective cohort of 231 colorectal cancer patients. The deep learning system was developed via migration learning for colorectal cancer detection, staging, and RAS gene mutation status prediction. This study used pre-trained Yolov7, vision transformer (VIT), swin transformer (SWT), EfficientNetV2, and ConvNeXt. 4620, and contrast-enhanced CT images and annotated tumor bounding boxes were included in the tumor identification and staging dataset. A total of 19,700 contrast-enhanced CT images comprise the RAS gene mutation status prediction dataset. Results: In the validation cohort, the Yolov7-based detection model detected and staged tumors with a mean accuracy precision (IoU = 0.5) (mAP_0.5) of 0.98. The area under the receiver operating characteristic curve (AUC) in the test set and validation set for the VIT-based prediction model in predicting the mutation status of the RAS genes was 0.9591 and 0.9554, respectively. The detection network and prediction network of the deep learning system demonstrated great performance in explaining contrast-enhanced CT images. Conclusion: In this study, a deep learning system was created based on the foundation of contrast-enhanced CT portal-phase imaging to preoperatively predict the stage and RAS mutation status of colorectal cancer patients. This system will help clinicians choose the best treatment option to increase colorectal cancer patients’ chances of survival and quality of life.

## 1. Introduction

The most prevalent gastrointestinal tract cancer worldwide and the second-leading cause of tumor-related mortality is colorectal cancer [1]. In the past few decades, significant improvements have been made in the care of people with colorectal cancer [2]. In recent years, advances in treatment strategies have played a significant role in raising survival rates [3,4]. However, the overall survival of patients with advanced colorectal cancer remains poor. The choice of therapy options for people with colorectal cancer depends on TNM staging. Medical imaging is frequently used to preoperatively assess patient staging; nevertheless, its accuracy in determining TNM staging remains poor [5].

Several studies have reported the use of contrast-enhanced CT to predict patient outcomes through precise identification and stratification of patients carrying specific mutated genes [6,7]. Tumor genetic profiling is a powerful tool to personalize therapy with the creation of customized treatments. All patients with suspected or confirmed metastatic colorectal cancer should be genotyped for tumor tissue RAS mutations, as these mutations predict resistance to the anti-epidermal growth factor receptor (EGFR) monoclonal antibodies, cetuximab and panitumumab. This recommendation is made in clinical guidelines on a regular basis [8,9]. Therefore, identification of RAS mutation status before or during treatment is essential for predicting treatment outcomes and determining individualized treatment strategies for colorectal cancer patients. In clinical practice, biopsies or postoperative collections are the most often utilized genetic testing techniques. Due to the intra-tumoral heterogeneity of colorectal cancer, these procedures are invasive, and local tumor sampling and biopsy techniques might not be representative [10].

In oncology, the accurate identification of imaging biomarkers is critical to enable clinicians to individualize their treatment choices [11]. According to previous studies, medical imaging can capture tumor biology at the genetic and cellular levels [12]. A common imaging test for patients with colorectal cancer who are being evaluated before surgery is contrast-enhanced CT, which is widely utilized in clinical settings [13]. Deep learning has recently achieved close attention in oncology research due to its ability to extract more information from input data [14,15,16,17]. The predictive performance of deep learning models under specific conditions has been demonstrated to be no worse than that of experienced clinicians [18,19]. This provided the basis for our study.

Thus, the objective of this investigation was to create a deep learning system for preoperative staging and RAS mutation status prediction in patients with colorectal cancer, which, to the best of our knowledge, has not been reported in any published studies.

## 2. Materials and Methods

Approval for this study was received from the Ethics Committee of the Second Affiliated Hospital of Nanjing Medical University (NO. 2023-KY-141-01). The patients’ or their family members’ informed consent was acquired. We removed all private patient information.

### 2.1. Patients

A total of 231 colorectal cancer patients took part in this study, which enrolled patients from January 2017 to June 2022. Patient staging and RAS mutation status were derived from postoperative pathological results. The inclusion criteria of this study were that contrast-enhanced CT examination was performed within one week before colorectal resection, postoperative pathology confirmed colorectal cancer, detection of RAS gene mutation status after colorectal resection and definite RAS gene mutation status, and no chemotherapy or radiotherapy before operation. The exclusion criteria of this study were poor gastric distension or artefacts in CT images, preoperative radiotherapy or chemotherapy, small colorectal cancer lesions that were difficult to identify, and being unable to determine RAS gene mutation status in the patient. The Appendix A provide detailed information on the testing methods for detecting RAS gene mutations in patients.

### 2.2. CT Image Acquisition

CT examinations were performed using a Siemens Definition Flash Dual Source CT (Somatom Definition, Siemens Healthcare, Forchheim, Germany). All patients were instructed to fast for more than 8 h and to inject anisodamine 20 mg intravenously to avoid gastric motility. Additionally, all patients were asked to take 1000 mL of warm water orally to dilate the stomach before the examination and to hold their breath during the examination. After the non-enhanced abdominal CT scan, the patients were intravenously injected with 1.5 mL/kg of iodinated contrast medium (ioversol injection 320 mg I/mL, Jiangsu Hengrui Pharmaceuticals Co., Ltd., Lianyungang, China) at a flow rate of 3.0 mL/s via an automatic pump syringe. After the contrast agent injection started, and when the contrast agent concentration reached 100 Hu, the imaging taken after 20 s was the arterial phase, the imaging taken at 35 s after the arterial-phase imaging was the venous phase, and the imaging taken at 90 s after the venous-phase imaging was the delayed phase. The parameters of the CT scan were as follows: tube voltage of 120 kV, tube current of 150–300 mA, field of view of 30–50 cm, matrix 512 × 512, rotation time of 0·5 s, and pitch of 1.0; the images were reconstructed with section thicknesses of 2 mm.

### 2.3. CT Images Collection

Studies demonstrated that characteristics taken from contrast-enhanced CT portal images had superior colorectal cancer prediction accuracy [20,21]. Thus, we collected the contrast-enhanced CT portal-phase images of all patients and resampled them. The supplemental information includes a comprehensive description of how the CT scans were acquired. All of the patients’ contrast-enhanced CT portal-phase images were examined by two radiologists with a combined expertise of more than eight years in medical imaging. They checked the quality of the patients’ enhanced CT images and screened the five images from each patient with the largest tumor area. The two physicians had no information about the patients’ pathology and their review process was independent of each other. If their opinions diverged, the final decision was made by a chief physician with 15 years of expertise in medical imaging.

### 2.4. Dataset Construction

From the CT images of each patient, to create the dataset, we chose five axial slices that had the greatest tumor area screened by the radiologists. Specifically, the section with the largest tumor cross-section was centered, with two sections above the center and two sections below, for a total of five sections (Figure 1).

We collected the patients’ pathologic staging from their postoperative pathology report. For automated tumor site recognition and prediction staging, we gathered 685 stage III images and 470 stage II images retrospectively. We expanded the original dataset via data augmentation. It decreased the likelihood that the model would be overfitting when processing the dataset [22]. After data enhancement, the dataset includes a total of 4620 images.

We retrospectively collected 525 CT images with gene mutations and 460 CT images without gene mutations for gene mutation status prediction in colorectal cancer patients. We expanded the dataset by applying 19 transformations to the original images using image enhancement techniques.

All of the images were first normalized. After that, these images were arbitrarily split into three groups: a training cohort, a testing cohort, and a validation cohort, in the order of (7:2:1). The training cohort was used to train the model, the test cohort was used for fine-tuning, and the validation cohort was used for evaluating the model’s effectiveness.

### 2.5. Model Construction

The construction of the model consists of two parts. Using the contrast-enhanced CT venous-phase images, the first step is to identify the tumor and determine the colorectal cancer stage (detection model). Predicting the tumor’s genetic status (prediction model) is the second step.

Using Yolov7, which was pre-trained on the Coco dataset, we created a detection model [23]. Data enhancement methods like HSV, translate, flip, scale, mix-up, and mosaic were used in the construction of detection model. We set the learning rate to 0.01 and the number of epochs to 200.

We preprocessed the images of the training cohort and the test cohort differently while creating the prediction model [24]. The data enhancement techniques of random clipping and random transformation were selected for the training cohort, and the data enhancement techniques of center cropping and resizing were selected for the test and validation cohorts. EfficientNetV2 and ConvNeXt achieve high accuracy in comparison to other models while utilizing fewer processing resources [25,26]. The transformer’s structure has high non-local feature extraction capabilities, and VIT and SWT perform well in the classification space [27,28,29]. All of them had prior training using the ImageNet dataset [30,31]. We set the learning rate of the convolutional neural network to 0.01, the learning rate of the transformer to 0.001, and the number of epochs to 200.

### 2.6. Model Evaluation

Three categories make up the YOLOv7 loss function: class loss, location loss, and objection loss. On each layer of the feature maps, the loss computation was performed. The precision–recall (P-R) curves, mAP, confusion matrix, and F1 score curve were utilized to further assess the detection model’s performance.

For the prediction model, the accuracy and loss values were used to evaluate the classification performance of the neural network, from which the best neural network was selected. To assess the network’s performance further, receiver operator characteristic (ROC) curves and P-R curves were employed. The output provided to the neural layers was shown using gradient-weighted class activation mapping (Grad-CAM) [32].

## 3. Results

### 3.1. Patients

A total of 231 patients were included in the study. Table 1 provides a summary of the clinical characteristics of the patients included in the detection model dataset. Additionally, Table 2 presents the clinical characteristics of the patients included in the prediction model dataset.

### 3.2. Detection Model Performance

The Yolov7 found the best optimization settings after 180 learning epochs. In the test cohort, the detection model had a precision of 0.96, a recall of 0.95, and a mAP_0.5 of 0.97 (Figure 2). Figure 3A,B depict the confusion matrix for both the test and validation cohorts. The results showed that in both cohorts, the detection model performed excellent classification of both stages. Furthermore, we used the mAP and F1 scores to evaluate the accuracy of the model’s detection performance. The test cohorts’ and validation cohorts’ mAP_0.5 values were 0.981 and 0.970, respectively (Figure 3C,D). The model’s F1 scores in the test and validation cohorts were 0.95 and 0.96, respectively (Figure 3E,F). This demonstrated that the model performed well in terms of detection.

### 3.3. Prediction Model Performance

Based on the accuracy and loss of the neural network training process, all neural networks reached the optimal optimization parameters after 170 learning cycles (Figure 4). The results show that VIT has the best classification performance for our dataset (Figure 3). Therefore, we chose VIT to construct the prediction model. The confusion matrix findings revealed that the prediction model performed well in both the test and validation cohorts (Figure 5A,B). Additionally, the ROC curve and P-R curve show that the prediction model has excellent classification performance in the validation cohort (Figure 5C,D). The AUC for the test and validation cohorts are 0.9591 and 0.9554, respectively (Delong Test, *p* = 0.449). The representative images of VIT’s Grad-CAM are shown in Figure 6. Based on Grad-CAM, we selected the first Layer Norm layer in the last Encoder Block module in VIT to generate a representative image of VIT. The heatmap generated based on VIT shows important regions in the CT image. The surrounding area and central position of the tumor in the CT image are of great value for the evaluation of tumor gene status. This suggests that VIT has the ability to detect tumor heterogeneity.

### 3.4. Deep Learning System

The deep learning system is made up of two parts: the detection model detects tumors and predicts staging, and the prediction model predicts RAS gene mutation status.

## 4. Discussion

We developed and validated an upgraded computed tomography-based deep learning system for preoperative prediction of staging and RAS gene mutation status in colorectal cancer patients in this work. The deep learning system successfully differentiated colorectal cancer patients based on staging and RAS gene mutation status, allowing tailored preoperative staging and RAS gene mutation status evaluation.

Accurate staging evaluation and RAS gene mutation status identification are critical in colorectal cancer patients’ therapy options and prognosis assessment [33]. Medical imaging is a typical tool for determining preoperative staging, although its accuracy is not optimal [34]. Endoscopic biopsy is a typical preoperative approach for determining gene mutation status. It can, however, result in major problems such as infection, bleeding, and perforation [35]. Several studies have sought to evaluate RAS gene mutation status using positron emission tomography (PET/CT) [36]. Although some results have been achieved, this is not a standard preoperative test for colorectal cancer patients. Contrast-enhanced CT is more commonly used for tumor detection and treatment [37]. CT imaging involves the scanning of a certain thickness of a layer of the body’s examination area using an X-ray beam. X-rays transmitted through the layer are received by a detector, converted into visible light, and then transformed into electrical signals by an optical-to-electrical converter, and then into digital signals by an analog/digital converter, which are inputted into a computer for processing. This contains a great deal of information. It has been shown that deep learning features extracted from a tumor region can offer a relevant quantitative representation of the extent of lymph node metastasis in patients [38]. Deep learning can extract more information from the input data for mapping the input data or for observing the relationship between features and the output and is not dependent on understanding the features of the data [14]. Thus, deep learning will mine information from medical CT images that is difficult for humans to notice, increasing the hope of achieving diagnostic value. To the best of our knowledge, this is the first study to employ contrast-enhanced CT imaging with deep learning to predict staging and RAS gene mutation status before surgery in colorectal cancer.

The bulk of deep learning research in colorectal cancer has focused on categorization of endoscopic or pathological images, diagnosis, and prognosis analysis [39,40]. There are many studies that attempted to use deep learning to analyze pathological images of colorectal cancer patients to predict lymph node status of patients [41]. Some researchers have also investigated the direct use of endoscopic images to diagnose the depth of submucosal infiltration in colorectal cancer using deep learning [42]. After a systematic literature search, we found two deep learning studies using magnetic resonance imaging (MRI) for the predictive identification of the T-stage in rectal cancer patients [43,44]. Although MRI has advantages such as high sensitivity and specificity, it is not a routine preoperative test for colorectal cancer patients due to its high price. Moreover, we did not find studies that used CT images to predict the stage of patients with colorectal cancer preoperatively. Clinical standards demand that practitioners identify TNM staging before beginning any therapy [45]. TNM staging is routinely used for risk stratification and therapeutic decision making, and CT is a routine imaging test used for preoperative staging of gastric cancer [46]. The use of abdominal contrast-enhanced CT has greatly improved the accuracy of gastric cancer staging, with preoperative T-stage and N-stage accuracies of 70% and 75%, respectively [47,48]. However, the final interpretation of CT images still depends on clinical experience and the personal opinion of radiologists. The staging assessment by clinicians is to some extent a subjective evaluation that lacks objectivity [49]. The F1 value of the deep learning system constructed in this study reached 0.95 and 0.96 in the test cohort and validation cohort, respectively, offering a novel approach for assisting radiologists in screening and reducing their workload. Huang et al. demonstrated that combining numerous indicators into a single model aids in individualizing patient care and outperforms utilizing a single marker [21]. We are strongly inclined toward the view that focusing on the T-stage or N-stage may not allow a thorough assessment of patients’ state, which may alter clinician’s diagnosis, treatment, and prognosis appraisal of the patients. Using contrast-enhanced CT portal-phase images, the detection model we constructed was able to predict the staging of colorectal cancer patients, which would be helpful to clinicians in making their diagnosis and treatment decisions.

With the introduction of cetuximab and panitumumab, two anti-epidermal growth factor receptor (EGFR)-targeted antibodies, the treatment of progressive colorectal cancer has entered the era of customized therapy [50]. However, tissues obtained via endoscopic biopsy are inaccurate, and approximately one-quarter of patients diagnosed with endoscopic biopsy prove to have more advanced disease after surgical resection [51]. Endoscopic biopsy specimens’ gene expression profiles may be influenced by sampling mistakes [52]. Liquid biopsy has evolved in recent years as an alternate approach for identifying genetic status [53]. However, the pricey apparatus needed for the investigation, the long analysis time, and the low detectability and specificity have all hampered the translation of this innovative technology from the laboratory to clinical use [54]. Contrast-enhanced CT is a relatively low-risk, non-invasive preoperative routine scan compared to endoscopy and tissue biopsy [55]. In this study, a prediction model was built and tested on a test set and a validation set, and the model achieved good performance. Our findings revealed that contrast-enhanced CT, as a typical preoperative scan used in colorectal cancer patients, has intrinsic receptor expression features and can thus represent RAS gene mutation status. Several studies have shown links between CT features and genes in lung tumors [56]. Predictive models are not yet sufficient to replace pathology biopsies for a variety of reasons, including clinician bias and poor deep learning interpretability. However, deep learning and contrast-enhanced CT examinations have many advantages over pathology biopsies. First, CT examinations are readily available, relatively inexpensive, and noninvasive. In addition, almost all patients with colorectal cancer undergo contrast-enhanced CT before treatment and are commonly imaged multiple times during treatment, but not all patients undergo genomic sequencing. Second, colorectal cancer is highly heterogeneous and progressive at the physiological and genomic levels. Genomic heterogeneity in different locations of primary tumors and metastatic tumors is a significant contributor to treatment failure and the establishment of therapeutic resistance. Third, when genomic analysis is performed, tumor biopsy samples are obtained from a single location in a single pass, which is prone to sampling errors. However, predictive models target images of the entire tumor, and not a localized site [57,58]. Therefore, precise treatment of colorectal cancer requires spatiotemporal analysis of tumor RAS gene mutation status. The findings of this study highlight the fact that contrast-enhanced CT in colorectal cancer has an intrinsic advantage for detecting RAS gene mutation status. This is useful since contrast-enhanced CT makes it easier for doctors to establish the mutational status of genes. Grad-CAM can depict the deep learning model’s output, and further research should be undertaken based on this finding. Contrast-enhanced CT and deep learning have the ability to quantify intra- and inter-tumor heterogeneity and enable more accurate colorectal cancer therapy.

Radiomics is a new topic that has attracted a lot of interest in cancer clinical research [59]. With the greatest AUC value of 0.76 obtained, Li et al. created clinical line graphs based on radiomic characteristics for predicting lymph node metastases in colorectal cancer patients [60]. The combination of functional parameters of CT and radiomic features is helpful for the diagnosis and T-staging of colorectal cancer [61]. Xue et al. used radiomic features to construct a model to predict KRAS mutation status in colorectal cancer patients, with an AUC value of 0.75 [62].

This study’s prediction model had a much higher AUC than its radiomic equivalent. This study’s remarkable result can be credited to the usage of deep learning techniques. Yun et al. discovered that combining deep learning features with radiomic features affects their deep learning model’s classification performance [63]. According to Chalkidou et al., radiomic properties are characterized by human bias [64]. Simultaneously, radiomics has always had repeatability issues [65]. The usefulness of classical radiomics has been called into doubt since the introduction of deep learning [66,67]. Deep learning enables important properties to be learnt automatically, without researchers’ prior definition, and these abstract representations also improve learning by boosting generality and accuracy while minimizing possible bias [68]. We tend to favor this view. The distinctions between different tissue types might not be adequately accounted for when examining radiomic characteristics due to the constraints of human-defined radiomics.

More importantly, this study looks at the therapeutic use of deep learning models. Previous deep learning research results, while good, have only been evaluated using internal or external test groups and have not been applied to clinical practice, which is contrary to the trend of personalized medicine [69,70,71]. Schmidt et al. contend that medical research should be directed toward therapeutic applications [72]. Thus, a deep learning system for clinicians based on the best model is useful, and our model demonstrates excellent predictive performance. Without specialist annotation, as clinicians upload contrast-enhanced CT images obtained from colorectal cancer patients, the proposed deep learning system displays summary results for patient staging and RAS gene mutation status prediction. Despite the obstacles in transferring medical research findings into the development of therapeutic technologies, as Cabitza et al. pointed out, we feel it is a worthy undertaking [73,74]. With its rapid learning and data processing capabilities, deep learning will revolutionize how we respond to colorectal cancer and become a vital tool for physicians.

This study has a number of drawbacks. To begin with, the colorectal cancer patients in this study were recruited from a single location, and the deep learning system may perform poorly on contrast-enhanced CT scans from other institutions. We will make a deliberate effort in future research to eliminate variance between hospitals using multi-center studies and significantly improve the deep learning system. Furthermore, only contrast-enhanced CT portal-phase images were employed in this investigation for prediction. The use of arterial-phase and delayed-phase contrast-enhanced CT imaging in colorectal cancer has to be researched further. Finally, a two-dimensional model was used to build the deep learning system for this investigation. We will investigate the clinical use of 3D models in contrast-enhanced CT.

## 5. Conclusions

In conclusion, the proposed deep learning system can predict the preoperative staging and RAS gene mutation status of colorectal cancer patients using just contrast-enhanced CT images. The deep learning system will assist physicians in evaluating the staging and RAS mutation status of colorectal cancer patients prior to surgery and selecting the best treatment strategy, thus decreasing the physical and financial stresses on patients.

## Figures and Tables

**Figure 1 cancers-15-04497-f001:**
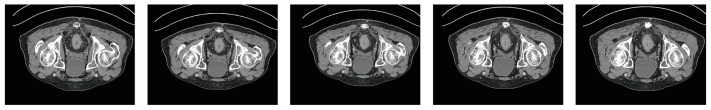
Examples of images selected for this study.

**Figure 2 cancers-15-04497-f002:**
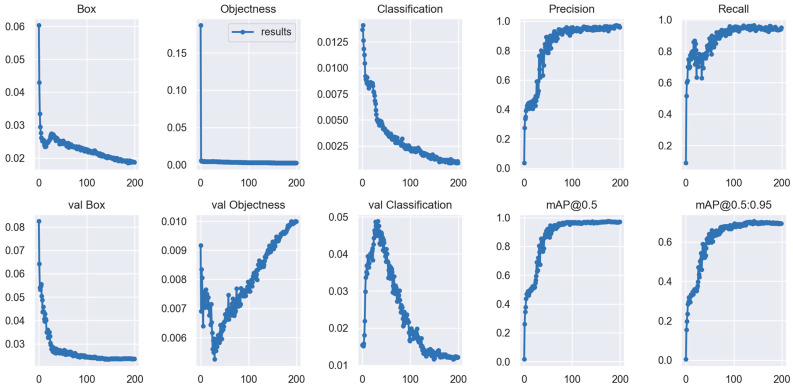
Variation in each metric using the Yolov7 over 200 epochs. mAP_0.5: mean average precision (IoU = 0.5). The Yolov7 attained the best-optimized parameters after 180 learning epochs, with a precision of 0.96 and a recall of 0.95 in the test cohort.

**Figure 3 cancers-15-04497-f003:**
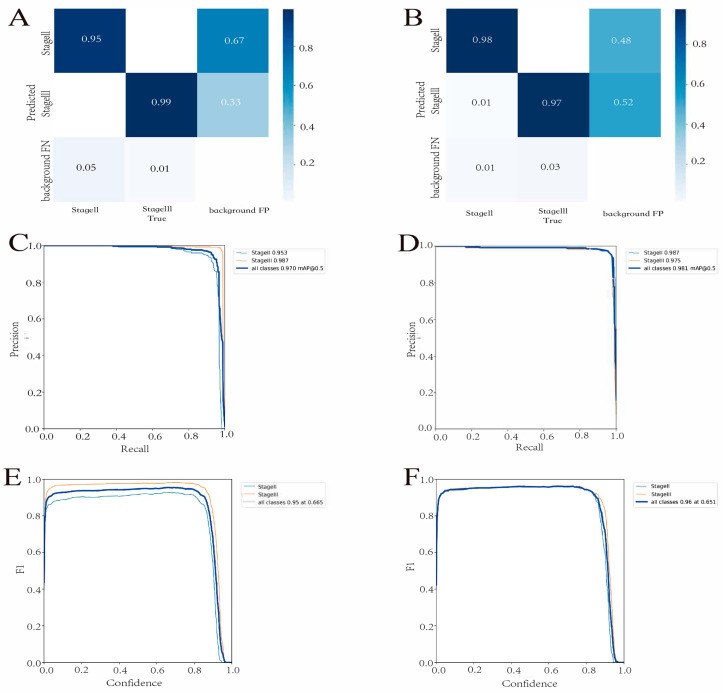
Performance of the detection model in the test and validation cohorts. (**A**,**B**) The confusion matrix in the test cohort (**A**) and the validation cohort (**B**). (**C**,**D**) The P-R relationship. The test cohort’s (**C**) and external validation cohort’s (**D**) mAP_0.5 values are 0.981 and 0.970, respectively. (**E**,**F**) The F1 curve. The model’s F1 scores in the test (**E**) and validation (**F**) cohorts are 0.95 and 0.96, respectively.

**Figure 4 cancers-15-04497-f004:**
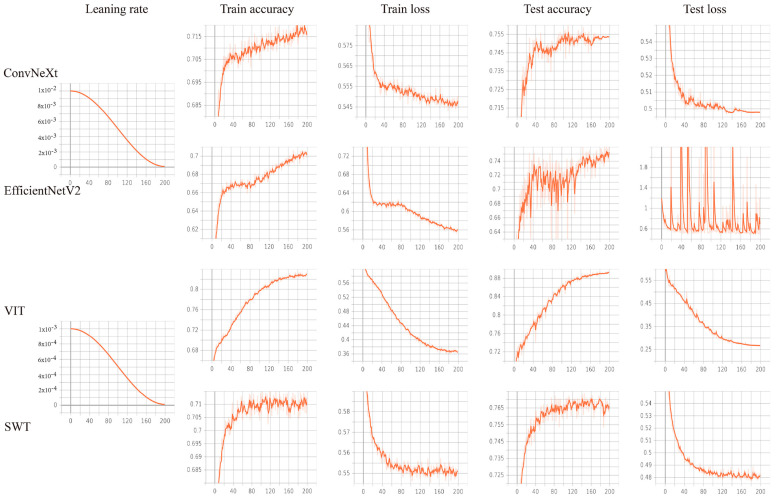
Variation in each metric over 200 epochs using different neural networks. After 170 learning epochs, all neural networks attained the best-optimized parameters based on the training loss and accuracy value. In the test cohort, the VIT model outperformed the CNNs in classification.

**Figure 5 cancers-15-04497-f005:**
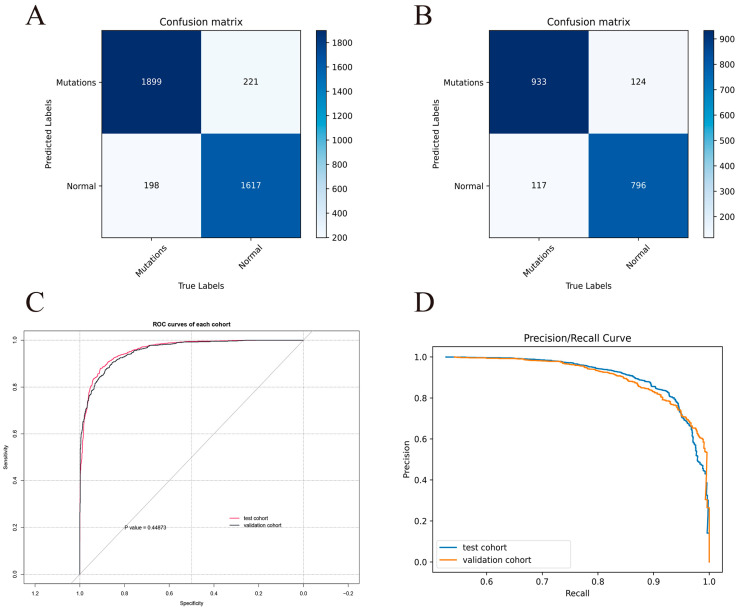
Evaluation of the prediction model’s performance. (**A**,**B**) The confusion matrix in the test (**A**) and external validation (**B**) cohort. (**C**) The ROC curves. The AUC values of test cohort and external validation cohort are 0.9591 and 0.9554, respectively. (**D**) The P-R curves. These results show that the prediction model has good classification performance.

**Figure 6 cancers-15-04497-f006:**
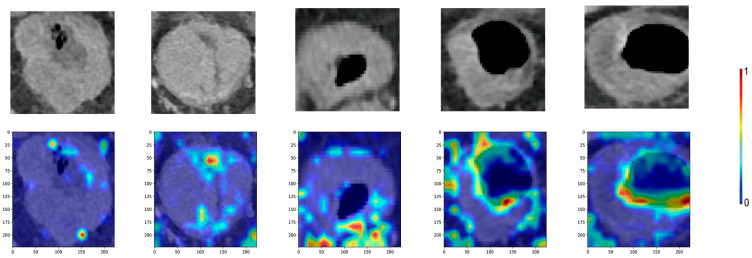
The prediction model provided enhanced CT portal-phase images and a feature heat map. Color bars show the prominence of the characteristics.

**Table 1 cancers-15-04497-t001:** Characteristics of patients included in the detection model.

Clinical Characteristics		
Age (mean ± SD)		63.97 ± 11.086
Gender, NO (%)		
	Male	170
	Female	61
Laboratory tests, median (IQR)		
	Albumin	40.60 (37.30, 42.90)
	Neutrophil	4.49 (3.23, 6.52)
	Lymphocyte	1.29 (0.94, 1.71)
CEA level, NO (%)		
	Normal	179
	Abnormal	52
CA125 level, NO (%)		
	Normal	203
	Abnormal	28
CA199 level, NO (%)		
	Normal	186

**Table 2 cancers-15-04497-t002:** Characteristics of patients included in the prediction model.

Clinical Characteristics		
Age (mean ± SD)		63.79 ± 11.143
Gender, NO (%)		
	Male	148
	Female	49
Laboratory tests, median (IQR)		
	Albumin	40.50 (37.85, 43.65)
	Neutrophil	4.11 (3.15, 5.86)
	Lymphocyte	1.34 (1.05, 1.74)
CEA level, NO (%)		
	Normal	145
	Abnormal	52
CA125 level, NO (%)		
	Normal	165
	Abnormal	32
CA199 level, NO (%)		
	Normal	158

## Data Availability

The data supporting this study are available from the corresponding author upon request.

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
