# Peer review of "A Contrast-Enhanced CT-Based Deep Learning System for Preoperative Prediction of Colorectal Cancer Staging and RAS Mutation"

_cancers, 2023, doi:10.3390/cancers15184497_

Round 1
Reviewer 1 Report
The authors developed a contrast enhanced CT-based deep learning system for preoperative prediction of colorectal cancer staging and RAS mutation with high accuracy in the validation cohort. The authors included 231 colorectal cancer patients. Their conclusion was that a deep learning system was created based on a contrast enhanced CT portal phase to preoperatively predict the stage and RAS mutation. This system will help clinicians choose the best treatment option to increase colorectal cancer patients’ chances of survival and quality of life. However, there are a couple of issues to be addressed as follows.
Comments
1. The authors claimed that a contrast enhanced CT-based deep learning system could predict colorectal cancer staging. The authors chose 5 slices including tumor area. Can they discuss why CT images of the tumor area can predict lymph node metastasis?
2. The authors described that medical imaging is atypical tool for determing preoperative staging, but its accuracy is not optimal. Can they discuss the comparison of the accuracy of predictions using medical imaging vs. predictions using CT-based deep learning system?
3. On the prediction of RAS mutation, do the authors believe that the predictive accuracy using CT-based deep leaning system is superior to biopsy diagnosis?
4. What happens when we tell patients that a colon cancer has a high likelihood of advanced stage and RAS mutation, as a result, they will need additional treatment including chemotherapy because of deep learning system that we don’t fully understand?
Author Response
Dear Reviewer1,
Thank you for the opportunity to resubmit our revised manuscript. We thank you for your constructive suggestions and comments. I am so sorry to bring you so much trouble because of our carelessness. Our point-by-point answers to your suggestions are below. We have revised the manuscript. The revised parts are highlighted in yellow.
Thanks very much again for your attention to our manuscript. Once again, thank you for your help to our manuscript processing.
We hope that the revised manuscript is now acceptable for publication. We look forward to hearing from you.
Sincerely,
Jianping Zhang
For your guidance is appended below.
Reviewer 1:
Question 1: The authors claimed that a contrast enhanced CT-based deep learning system could predict colorectal cancer staging. The authors chose 5 slices including tumor area. Can they discuss why CT images of the tumor area can predict lymph node metastasis?
Response: We are grateful for this advice.
CT imaging is the scanning of a certain thickness of a layer of the body's examination area with an X-ray beam. X-rays transmitted through the layer are received by a detector, converted into visible light, and then transformed into electrical signals by an optical-to-electrical converter, and then into digital signals by an analog/digital converter, which are inputted into a computer for processing. This contains a great deal of information. It has been shown that deep learning features extracted from the tumor region can reflect a relevant quantitative representation of the extent of lymph node metastasis in patients[1]. Deep learning can extract more information from the input data for mapping the input or observing the relationship between features and the output and is not dependent on understanding the features of the data[2]. Thus, deep learning will mine information from medical CT images that is difficult for humans to notice, increasing the reasonable hope of diagnostic value. We have added them in the Discussion.
Question 2: The authors described that medical imaging is atypical tool for determing preoperative staging, but its accuracy is not optimal. Can they discuss the comparison of the accuracy of predictions using medical imaging vs. predictions using CT-based deep learning system?
Response: Thank you very much for your suggestion.
TNM staging is routinely used for risk stratification and therapeutic decision making, and CT is a routine imaging test used for preoperative staging of gastric cancer[3]. The use of abdominal-enhanced CT has greatly improved the accuracy of gastric cancer staging, with preoperative T-staging and N-staging accuracies of 62-75% and 75- 80%, respectively[4, 5]. However, the final interpretation of CT images still depends on clinical experience and the personal opinion of the radiologist. The staging assessment by clinicians is to some extent a subjective evaluation that lacks objectivity[6]. The F1 value of the deep learning system constructed in this study reached 0.95 and 0.96 in the test cohort and validation cohort, respectively, and is a novel approach to in assisting radiologists in screening and reducing their workload. We have added them in the Discussion.
Question 3: On the prediction of RAS mutation, do the authors believe that the predictive accuracy using CT-based deep leaning system is superior to biopsy diagnosis?
Response: Many thanks for the advice.
Recently, pathologic diagnosis is the gold standard for disease diagnosis, and the deep learning system is also trained based on the results of molecular pathology of patients, so we do not believe that the deep learning system we constructed is superior to biopsy diagnosis in the prediction of RAS gene mutations. Deep learning can extract more information from the input data for mapping the input or observing the relationship between features and the output and is not dependent on understanding the features of the data. We think that the deep learning system may be of greater value for clinical diagnosis and treatment under conditions where the quality of endoscopic biopsies is poor. In addition, the purpose of the deep learning system constructed in this study is aimed at assisting clinicians in clinical diagnosis and treatment decision making, and is not intended to replace pathological biopsy.
Question 4: What happens when we tell patients that a colon cancer has a high likelihood of advanced stage and RAS mutation, as a result, they will need additional treatment including chemotherapy because of deep learning system that we don’t fully understand?
Response: Many thanks for these suggestions.
Because the patient's colorectal cancer is in advanced stage, according to the clinical guidelines, neoadjuvant therapy should be performed first to improve the patient's stage, and then a systematic evaluation should be performed before deciding whether the patient can undergo surgical treatment. Patients with RAS gene mutation will be resistant to cetuximab and panitumumab, and then chemotherapy regimens that include these two targeted drugs should be avoided in the process of treatment plan development. In the past, the clinician may find that the patient is inoperable during surgical exploration or may choose a treatment plan that includes these two targeted agents, which may increase the patient's pain, delay treatment and lead to disease progression, and increase the cost of treatment.
Thanks again for your comments!
Reference
(1) Dong, D.; Fang, M.-J.; Tang, L.; Shan, X.-H.; Gao, J.-B.; Giganti, F.; Wang, R.-P.; Chen, X.; Wang, X.-X.; Palumbo, D.; Fu, J.; Li, W.-C.; Li, J.; Zhong, L.-Z.; Cobelli, F. D.; Ji, J.-F.; Liu, Z.-Y.; Tian, J. Deep Learning Radiomic Nomogram Can Predict the Number of Lymph Node Metastasis in Locally Advanced Gastric Cancer: An International Multicenter Study. Ann. Oncol. 2020, 31 (7), 912–920. https://doi.org/10.1016/j.annonc.2020.04.003.
(2) S, H.; J, Y.; S, F.; Q, Z. Artificial Intelligence in Cancer Diagnosis and Prognosis: Opportunities and Challenges. Cancer Lett. 2020, 471. https://doi.org/10.1016/j.canlet.2019.12.007.
(3) Lu, Y.; Yu, Q.; Gao, Y.; Zhou, Y.; Liu, G.; Dong, Q.; Ma, J.; Ding, L.; Yao, H.; Zhang, Z.; Xiao, G.; An, Q.; Wang, G.; Xi, J.; Yuan, W.; Lian, Y.; Zhang, D.; Zhao, C.; Yao, Q.; Liu, W.; Zhou, X.; Liu, S.; Wu, Q.; Xu, W.; Zhang, J.; Wang, D.; Sun, Z.; Gao, Y.; Zhang, X.; Hu, J.; Zhang, M.; Wang, G.; Zheng, X.; Wang, L.; Zhao, J.; Yang, S. Identification of Metastatic Lymph Nodes in MR Imaging with Faster Region-Based Convolutional Neural Networks. Cancer Res. 2018, 78 (17), 5135–5143. https://doi.org/10.1158/0008-5472.CAN-18-0494.
(4) Kubota, K.; Suzuki, A.; Shiozaki, H.; Wada, T.; Kyosaka, T.; Kishida, A. Accuracy of Multidetector-Row Computed Tomography in the Preoperative Diagnosis of Lymph Node Metastasis in Patients with Gastric Cancer. Gastrointest. Tumors 2017, 3 (3–4), 163–170. https://doi.org/10.1159/000454923.
(5) Joo, I.; Lee, J. M.; Kim, J. H.; Shin, C.-I.; Han, J. K.; Choi, B. I. Prospective Comparison of 3T MRI with Diffusion-Weighted Imaging and MDCT for the Preoperative TNM Staging of Gastric Cancer. J. Magn. Reson. Imaging JMRI 2015, 41 (3), 814–821. https://doi.org/10.1002/jmri.24586.
(6) Zheng, L.; Zhang, X.; Hu, J.; Gao, Y.; Zhang, X.; Zhang, M.; Li, S.; Zhou, X.; Niu, T.; Lu, Y.; Wang, D. Establishment and Applicability of a Diagnostic System for Advanced Gastric Cancer T Staging Based on a Faster Region-Based Convolutional Neural Network. Front. Oncol. 2020, 10, 1238. https://doi.org/10.3389/fonc.2020.01238.

Reviewer 2 Report
Interesting and up-tp-date topic of research
In the file in attachement, the authors can find the recommended modifications
-particularly, avoid the use of the terms photos or pictures with referral to radiological images
Major concerns:
-inclusion/exclusion criteria should be placed in the manuscript not in the suppl materials
-the selection of the slices with the greater tumor area should be better explained, also with a figure
- CT acquisition and reconstruction protocol should be explained clearly and in a detail way, as it has a direct impact on the reliability on the study. Moreover, the equipment of CT performing should be specified, as well as the modality of contrast medium administration
Minor changes in the attached file

Author Response
Dear Reviewer2,
Thank you for the opportunity to resubmit our revised manuscript. We thank you for your constructive suggestions and comments. I am so sorry to bring you so much trouble because of our carelessness. Our point-by-point answers to your suggestions are below. We have revised the manuscript. The revised parts are highlighted in yellow.
Thanks very much again for your attention to our manuscript. Once again, thank you for your help to our manuscript processing.
We hope that the revised manuscript is now acceptable for publication. We look forward to hearing from you.
Sincerely,
Jianping Zhang
For your guidance is appended below.
Reviewer 2:
Question 1: avoid the use of the terms photos or pictures with referral to radiological images.
Response: We are grateful for this advice. We have made all the suggested changes as given in the pdf file.
Question 2: inclusion/exclusion criteria should be placed in the manuscript not in the suppl materials.
Response: We are grateful for this advice. We have added them to the Method.
Question 3: the selection of the slices with the greater tumor area should be better explained, also with a figure.
Response: We are grateful for this advice. We chose the images of a patient as an example for demonstration, as detailed in Figure 1.
Figure 1. Examples of images selected for this study.
Question 4: CT acquisition and reconstruction protocol should be explained clearly and in a detail way, as it has a direct impact on the reliability on the study. Moreover, the equipment of CT performing should be specified, as well as the modality of contrast medium administration.
Response: We are grateful for this suggestion.
2.2. CT image acquisition
CT examinations were performed on a Siemens Definition Flash Dual Source CT (Somatom Definition, Siemens Healthcare, Forchheim, Germany). The patient was in-structed to fast for more than 8 hours and to inject anisodamine 20 mg intravenously to avoid gastric motility. Besides, all patients were asked to take 1000ml of warm water orally to dilate the stomach before the examination and hold their breath during the examination. After the non-enhanced abdominal CT scan, the patients were intrave-nously injected with 1.5 mL/kg of iodinated contrast medium (ioversol injection 320 mg I/mL, Jiangsu Hengrui Pharmaceuticals Co.,Ltd, Jiangsu, China) at a flow rate of 3.0 mL/s by an automatic pump syringe. After the contrast agent injection starts, when the contrast agent concentration reached 100 Hu, the imaging after 20 seconds is the arterial phase, the imaging at 35 seconds after the arterial phase imaging is the venous phase, and the imaging at 90 seconds after the venous phase imaging is the delayed phase. The parameters of the CT scan were as follows: tube voltage 120 kV, tube cur-rent 150 - 300 mA, field of view 30 - 50 cm, matrix 512 × 512, rotation time 0·5 seconds, pitch 1.0, and images were reconstructed with section thicknesses of 2 mm. We have added them to the Method.
Question 5: Minor changes in the attached file.
Response: We are grateful for this suggestion. We have made all the suggested changes as given in the pdf file.
Thanks again for your comments!

Round 2
Reviewer 1 Report
The authors have revised the paper sufficiently.
Reviewer 2 Report
I thank the authors for performing the suggested revisions.
According to my opinion, the manuscript is suitable for publication